# FaceGPT: Self-supervised Learning to Chat about 3D Human Faces

## Abstract

We introduce FaceGPT, a self-supervised learning framework for large vision-language models (VLMs) to reason about 3D human faces from images and text. Typical 3D face analysis algorithms are specialized and lack semantic reasoning capabilities. FaceGPT overcomes this limitation by embedding the parameters of a 3D morphable face model (3DMM) into the token space of a VLM, enabling the generation of 3D faces from both textual and visual inputs. FaceGPT is trained as a model-based autoencoder in a self-supervised manner from in-the-wild images. In particular, a dedicated face token is projected to 3DMM parameters and then rendered as a 2D face image to guide the self-supervised learning process through image-based reconstruction. Without relying on expensive 3D annotations, FaceGPT learns to generate 3D faces based on visual or textual inputs, achieving a competitive performance compared to methods that are specialized to each of these tasks. Most importantly, FaceGPT is able to leverage the world knowledge in VLMs to achieve semantic reasoning capabilities, allowing the model to perform *speculative generation* of 3D faces purely from subtle textual prompts that do not explicitly describe facial features. This opens a new way of generating 3D faces from subtle descriptions of emotions or general everyday situations.

## 1 Introduction

In this work, we address the problem of reasoning about 3D human faces from images and text. Related work on monocular 3D face reconstruction aims to estimate the parameters of a 3D morphable model Blanz & Vetter (1999); Tewari et al. (2017); Deng et al. (2019b); Feng et al. (2021a); Li et al. (2023) given 2D face images as input. However, these methods lack the capability to reason about faces from text input. Unlike these systems, humans can vividly imagine and even draw faces based solely on either face images or textual descriptions. Motivated by recent advances in large vision-Language models (VLMs) Liu et al. (2023b); Zhu et al. (2023), we aim to explore a path forward towards enabling VLMs to obtain an in-depth reasoning-based understanding of 3D faces.

To investigate this question, we present FaceGPT, a vision-language model with an intricate ability to reason about 3D human faces from visual and textual input. We represent faces as 3D morphable model (3DMM) parameters that include parameters for the 3D face shape, expression, albedo, and scene illumination. Following related work on image segmentation Lai et al. (2024) and human pose estimation Feng et al. (2024), we extend the original token space of the VLM by incorporating a new <FACE> token that is decoded into 3DMM parameters using an MLP (Fig. 1). Thus, when combined with a differentiable computer graphics renderer Ravi et al. (2020), the VLM model becomes capable of synthesizing face images. This enables us to formulate FaceGPT within a model-based autoencoder framework Tewari et al. (2017), and hence to train our model in a fully self-supervised manner from in-the-wild images. To the best of our knowledge, this is the first work combining vision-language model with an inverse graphics pipeline. During training, we freeze the visual encoder of the VLM while training the MLP and the LLM using LoRA Hu et al. (2022). The model is trained with three types of data: (1) In-the-wild face images for the self-supervised training of the <FACE> token and 3DMM projection layers via inverse rendering. (2) Text-to-3DMM data for generating 3DMM parameters from text that explicitly describes facial features. (3) Standard multi-modal instruction tuning data to retain the general capability and quality of the VLM. We construct this dataset from a set of face images by running an off-the-shelf self-supervised monocular face reconstruction method Li et al. (2023) and by generating textual descriptions of the faces via the original VLM.

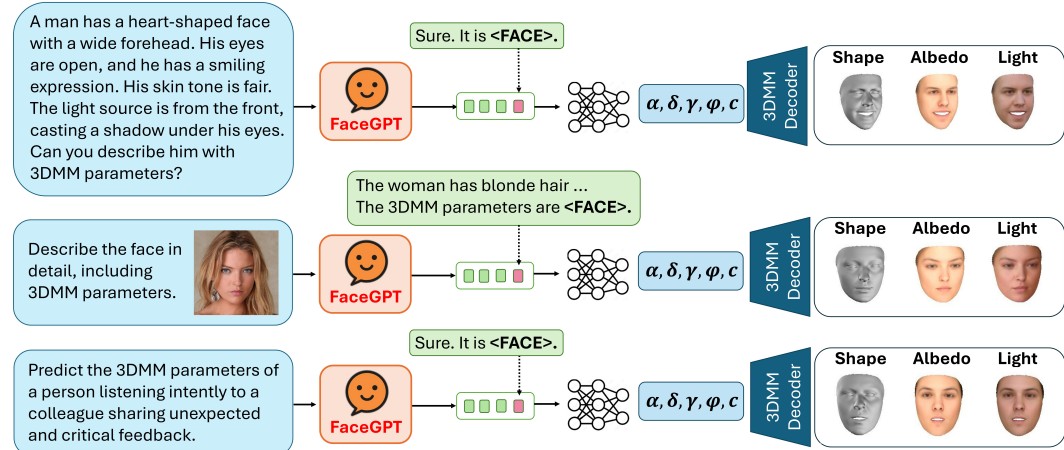

Figure 1: We introduce FaceGPT, a large vision-language model that learns to produce 3D human faces (in terms of 3DMM parameters) in a fully-self-supervised manner. When prompted with face images and task-specific questions, FaceGPT can output a special <FACE> token of which the corresponding feature embedding (red) can be decoded into 3DMM parameters, that encode the face shape $\alpha$, expression $\delta$, the texture $\gamma$, the light $\phi$ and camera $c$ parameters. When decoded with a 3DMM and differentiable renderer, this enables a fully self-supervised learning via inverse rendering. FaceGPT is a general-purpose model that can produce: 3D human faces from text-only input (first row), as well as from multi-modal input (second row). Moreover, FaceGPT is the first model capable of speculative face generation (last row), all while retaining general chatting abilities.

We evaluate FaceGPT on a variety of tasks, including text-to-3DMM face generation, traditional 3D face reconstruction, and general-purpose visual instruction following. We demonstrate that FaceGPT becomes a general-purpose model that achieves competitive results when compared to specialized methods in all those tasks. Most importantly, we show that FaceGPT is able to leverage the world knowledge in VLMs to achieve semantic reasoning capabilities, allowing the model to perform *speculative generation* of 3D faces purely from subtle textual prompts that do not explicitly describe facial features, such as "the person is listening intently to a colleague sharing unexpected and critical feedback" (Fig. 1). Hence, FaceGPT goes far beyond existing methods as it can translate implicit descriptions of emotional states into 3D facial features, which requires a semantic understanding of (i) how feelings like contemplation affect expressions and (ii) how these changes appear in realistic 3D facial features. Beyond face analysis, we believe that the design principles underlying FaceGPT are general and also suggest a pathway towards a self-supervised integration of the "world knowledge" that VLMs derive from extensive textual data and the structured 3D representations of the visual world via self-supervised learning through inverse rendering. In summary, our work makes the following concrete contributions:

- **A novel vision-language model (FaceGPT) for 3D face reasoning.** We propose the first approach to integrating 3D face understanding capabilities within a vision-language model (VLM). FaceGPT leverages both visual and textual inputs to reason about 3D facial geometry and appearance, bridging the gap between image-based 3D face reconstruction and text-based facial description interpretation.

- **Semantic reasoning for speculative 3D face generation.** Unlike traditional methods that rely on explicit visual or textual cues, FaceGPT demonstrates the ability to perform speculative 3D face generation based on abstract or emotional descriptions.

- **Competitive performance across multiple tasks.** We evaluate FaceGPT across various benchmarks, including 3D face reconstruction from images, text-to-3D face generation, and visual instruction following. Our results show that FaceGPT performs competitively with specialized 3D face reconstruction methods while maintaining the flexibility and reasoning capabilities of a general-purpose vision-language model.

The design principles of FaceGPT, specifically the integration of structured 3D representations with the world knowledge encoded in VLMs, provide a general framework for learning 3D-aware multimodal reasoning from in-the-wild 2D images in a self-supervised manner. Therefore, we believe FaceGPT represents a significant step forward in the field of vision-language models.

## 2 RELATED WORKS

### 2.1 MONOCULAR MODEL-BASED FACE RECONSTRUCTION

Realistically reconstructing digital human faces has been a longstanding challenge in computer vision and graphics due to their vast potential applications. Traditional methods primarily use parametric 3D Morphable Models (3DMM) Blanz & Vetter (1999); Paysan et al. (2009); Li et al. (2017) with PCA for dimensionality reduction to simplify high-dimensional 3D face scans, serving as a 3D prior for representing unique facial characteristics and providing precise control. Recently, deep learning-based methods that map 2D images to 3D face models have gained popularity. Early methods struggled with the need for extensive 3D facial scan data paired with 2D images, which was labor-intensive and costly. This limitation was addressed with the introduction of model-based face autoencoders (MoFA) Tewari et al. (2017) that enabled self-supervised 3D face reconstruction. MoFA uses a differentiable rendering layer to minimize differences between input and rendered images, enabling end-to-end learning without ground-truth 3D faces, leading to a number of effective extensions in the self-supervised learning strategies Tewari et al. (2018b;a); Bas et al. (2017); Genova et al. (2018); Daněček et al. (2022). RingNet Sanyal et al. (2019) and DECA Feng et al. (2021b) use landmark-based training, predicting landmarks for input images and treating them as pseudo ground truth, measuring the distance between 2D face landmarks and their projections on the 3DMM surface. The FOCUS Li et al. (2023) framework jointly trains a face autoencoder and an outlier segmentation network, which makes the method robust to outliers such as occlusion and make-up. These advancements significantly improved monocular model-based face reconstruction, making it more efficient and effective. However, these methods are highly specialized and lack a deep understanding of the semantics of human faces or the ability to relate faces to language, limiting their overall scope and effectiveness.

### 2.2 TEXT-TO-3D FACE GENERATION AND MANIPULATION

Text-to-3D face generation and manipulation methods aim to use textual information for creating and editing 3D faces. Methods like Dreamface Zhang et al. (2023) and Describe3D Wu et al. (2023b) generate text-conditioned texture maps to render 3D morphable models (3DMM). TG-3DFace Yu et al. (2023) advances this by using tri-plane neural representations and extending the 3D-aware GAN, EG3D Chan et al. (2022), for end-to-end text-conditioned generation. For text-guided 3D face manipulation, methods like Latent3D Canfes et al. (2022) and ClipFace Aneja et al. (2023) optimize intermediate layers with a CLIP-based loss to generate UV-texture maps or predict texture and expression latent codes. These methods, however, rely on 3D scan data and to train new mappers for each text instruction. Unlike these task-specific approaches, FaceGPT is a general-purpose model that reasons about 3D human faces from images, text, or both by leveraging general visual knowledge. Our model can interact with users through conversations, discussing facial features and providing relevant responses, while also being capable of following general user instructions.

### 2.3 MULTIMODAL LARGE LANGUAGE MODELS

Large Language Models (LLMs) are rapidly transforming various fields Radford et al. (2019); Brown et al. (2020); OpenAI (2024); OpenAI et al. (2024). While proprietary models like OpenAI's ChatGPT OpenAI (2024) and GPT-4 OpenAI et al. (2024) dominate the landscape, open-source alternatives such as Vicuna Chiang et al. (2023), LLaMA Touvron et al. (2023), Alpaca Taori et al. (2023), Mistral Jiang et al. (2023) and Qwen Bai et al. (2023) support research efforts. However, LLMs mainly focus on generating text as output given text-only input. The integration of additional modalities into LLMs represents an active area of research.

Multi-Modal Large Language Models (MM-LLMs) are emerging, extending LLMs' capabilities beyond text to encompass a broader spectrum of modalities, including images, videos, and audio. In the

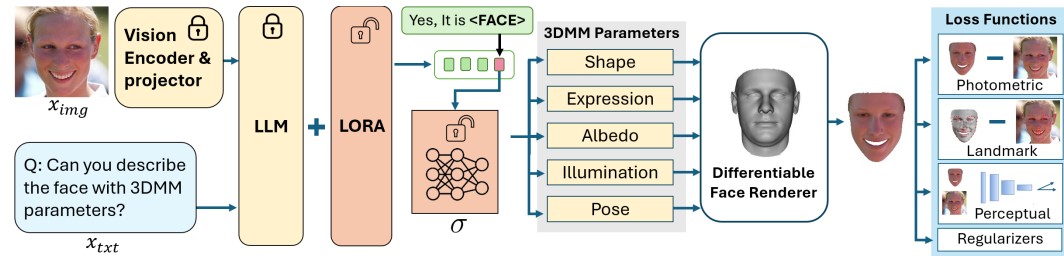

Figure 2: Architecture of FaceGPT. Our model consists of a vision-language model, which includes a vision encoder, a vision projection layer, and an LLM, along with a 3DMM projection layer, denoted as $\sigma$, and the parametric Basel face model Blanz & Vetter (1999). During training, the $\sigma$ projection layer is optimized and the LLM is fine-tuned through LORA, while keeping other components frozen. The training is guided through a self-supervised reconstruction loss using a differentiable renderer.

realm of image-text understanding, recent endeavors like LLaVA Liu et al. (2024) and MiniGPT-4 Zhu et al. (2023) incorporate vision encoders to interpret images and align their features with language embeddings using projection layers. Moreover, cutting-edge models such as PandaGPT Su et al. (2023), ImageBind Girdhar et al. (2023), and NeXT-GPT Wu et al. (2023c) exhibit impressive versatility in handling diverse modalities, aligning embeddings from text, images, audio, and video with language as both input and output. To enhance LLM to natively output more modalities, approaches like LISA Lai et al. (2024) connects LLaVA with a decoder to generate text and segmentation masks, while ChatPose Feng et al. (2024) specializes in human pose information. However, these methods typically rely on supervised learning.

Our aim is to develop a general-purpose vision-language model that is able to (1) generate 3D faces from text or image inputs and (2) capable of connecting the world knowledge from the LLM with 3D human faces to achieve semantic reasoning capabilites about faces in a self-supervised manner.

## 3 METHOD

In this work, our aim is to augment existing large Vision Language Models (VLMs) with the ability of reasoning about 3D human faces without requiring manual human efforts. Inspired by established model-based face reconstruction methods, we represent 3D human face with 3D Morphable Model (3DMM) Blanz & Vetter (1999) parameters, representing the *face shape*, *expression*, *albedo*, *illumination* and *pose*. In particular, we introduce a <FACE> token into the language space of the LLM, which is mapped into the 3DMM parameter space and subsequently rendered into a 2D image, hence enabling self-supervised 3D facial reconstruction. Fig. 2 presents the whole pipeline of FaceGPT.

### 3.1 MODEL ARCHITECTURE

**Representing Face in language space.** Inspired from recent advancements in LMMs, we treat the human face as a distinct modality and incorporate its representation into the language space of VLM. Specifically, we extend the vocabulary of VLM to include a new token <FACE> that specifically represents the human face. Given an input text prompt $\mathbf{x}_{txt}$ and/or input image $\mathbf{x}_{img}$, the VLM $f$ predicts text responses:

$$\mathbf{y}_{txt} = f(\mathbf{x}_{img}, \mathbf{x}_{txt}), \qquad (1)$$

where $\mathbf{y}_{txt} = [t_1, \ldots, t_N]$ is the output sequence of tokens with corresponding hidden states $[h_1, \ldots, h_N]$. When $\mathbf{x}_{txt}$ contains a face generation instruction, the resulting output $\mathbf{y}_{txt}$ will include a <FACE> token, facilitating further 3DMM predictions.

**From <FACE> token to 3DMM.** If one of the output tokens $t_n \in \mathbf{y}_{txt}$ is our defined <FACE> token, we can extract the hidden state as $h_{<FACE>} = h_n \in \mathbb{R}^{4096}$ and project it using an MLP $\sigma$ into the latent 3DMM parameters $\theta = \sigma(h_{<FACE>}) = [\alpha, \delta, \gamma, \phi, c] \in \mathbb{R}^{257}$, i.e. the 3D face shape $\alpha \in \mathbb{R}^{80}$, facial expression parameters $\delta \in \mathbb{R}^{64}$ and texture $\gamma \in \mathbb{R}^{80}$ of a 3DMM, as well as the spherical harmonics illumination $\phi \in \mathbb{R}^{27}$ and camera parameters $c \in \mathbb{R}^6$ of the scene. The 3D vertices and triangles, as well as the color of the face mesh are then determined using the standard 3DMM model

$M(\theta)$ as described in Tewari et al. (2017). Using an orthographic camera model the reconstructed 3D face mesh can be rendered into 2D space using a differentiable renderer $\Pi$, hence producing the final reconstructed face image $\hat{\mathbf{y}}_{rec}$. The process can be summarized as:

$$\theta = \sigma(h_{\texttt{<FACE>}}) \tag{2a}$$

$$\hat{\mathbf{y}}_{rec} = \Pi(M(\theta)) \tag{2b}$$

Note that the FaceGPT architecture can be seen as a new type of language-based autoencoder, with a VLM as encoder and a computer graphics decoder that is based on the 3DMM.

## 3.2 SELF-SUPERVISED TRAINING

Our objective is to develop a VLM that is able to learn an enhanced comprehension of 3D human faces in a self-supervised manner. Ultimately, the model should not only understand user instructions reliably, but also to accurately reconstruct 3D faces from visual or text input. To accomplish this, we have devised a self-supervised approach that incorporates 3DMM understanding into existing VLMs. This method allows us to leverage face data without the need for costly 3DMM annotations. We also construct a text-to-3DMM dataset in an unsupervised manner that supports this training paradigm, enabling our model to learn these capabilities effectively and efficiently.

**Self-supervised face reconstruction loss.** To incorporate the 3DMM as a new modality into an existing VLM, we add a new token <FACE> in the vocabulary of the LLM and fine-tune the language modelling head. For fine-tuning the VLM without relying on manually annotated data, we introduce a 2D self-supervised loss $L_{face}$, following established protocols of specialized face reconstruction models Li et al. (2023):

$$L_{face} = \lambda_{pixel}L_{pixel} + \lambda_{per}L_{perc} + \lambda_{LM}L_{LM} + \lambda_{reg}L_{reg}, \tag{3}$$

Specifically, $L_{face}$ incorporates the following components:

*Reconstruction loss.* $L_{pixel} = ||A \odot (x_{img} - \hat{y}_{rec})||_2^2$ refers to the pixel-wise reconstruction loss between reconstructed images $\hat{y}_{rec}$ and the input images $x_{img}$. To avoid distortions when training from in-the-wild data, a 2D skin mask $A$ is estimated from the input images using a simple pre-trained Gaussian mixture model for the skin color Deng et al. (2019b).

*Perceptual loss.* $L_{perc} = sim(f_{perc}(x_{img}), f_{perc}(\hat{y}_{rec}))$ estimates the cosine similarity between the reconstructed image $\hat{y}_{rec}$ and the input image $x_{img}$ at the perceptual level using a pre-trained feature extractor $f_{perc}$.

*Landmark loss.* $L_{LM} = ||LM_{img} - LM_{rec}||_2^2$ measures the L2 distance between the projected 2D landmarks of the estimated 3DMM $LM_{rec}$ and predicted 2D facial landmarks $LM_{img}$ using an off-the-shelf facial detector (Bulat & Tzimiropoulos, 2017).

*Parameter regularization.* $L_{reg} = ||\theta||_2^2$ regularizes the estimated 3DMM parameters towards the mean value of the multi-variate Gaussian distribution of the 3DMM.

The weight parameters $\lambda_{face} = [\lambda_{pixel}, \lambda_{per}, \lambda_{LM}, \lambda_{reg}]$ balance the respective losses, and we set them as established in prior work Li et al. (2023).

*Preserving the ability for natural conversations about faces.* We noticed that VLM's loose the ability to conduct general conversations about faces when trained self-supervised face reconstruction loss, and tend to always output 3DMM parameters when queried with a face image. To resolve this problem, we generate a face conversation dataset with accurate textual face descriptions, by querying the VLM with face images and conversations that include multiple questions about facial attributes. During training, we mix task-specific instructions that explicitly ask for 3DMM parameters with general conversational data to regularize the training and preserve the ability for general non-3DMM related conversations about faces. In contrast to prior works (Lai et al., 2024; Feng et al., 2024) which generally employ a simple static question-answer template that restrains the model from providing diverse responses, our proposed strategy can effectively improve the instruction following performance for general face conversations.

**Self-supervised Text-to-3DMM loss.** We aim to enable the VLM to perform semantic face understanding tasks, i.e. to predict faithful 3DMM parameters when only having textual input $f(\cdot, x_{txt})$.

While related works for 3D human pose estimation Delmas, Ginger and Weinzaepfel, Philippe and Lucas, Thomas and Moreno-Noguer, Francesc and Rogez, Grégory (2022); Feng et al. (2024) rely on human-written fine-grained language descriptions of 3D human meshes, we aim to achieve text-to-3DMM generation without any human interaction. To achieve this, we introduce a self-supervised text-based face reconstruction loss. In particular, we first query the VLM with the face images in the training data and a question template that requests detailed face descriptions of a number of facial attributes such as head shape, skin tone, or facial expressions . This results in a dataset with pairs of face images and detailed textual descriptions about facial attributes. During training, we instruct the VLM to estimate 3DMM parameters only from the available detailed face descriptions $f(\cdot, x_{txt})$. As we have the corresponding 2D image for each face description, we can re-use the self-supervised face reconstruction loss Eq. (3) to train the model at the task of text-to-3DMM generation, avoiding the need for any human-written annotation.

**Text prediction loss.** We also utilize the autoregressive objective $L_{txt}$ to guide the model to produce correct text output, thereby preserving its general ability to follow user instructions:

$$L_{txt} = CE(\hat{\mathbf{y}}_{txt}, \mathbf{y}_{txt}), \tag{4}$$

where $y_{txt}$ is the ground truth text response and $CE$ is the cross entropy loss.

**Overall training Objective.** Our overall training objective $L$ integrates both text-based autoregressive objective and a self-supervised loss for predicting 3DMM from either image or text input:

$$L = \lambda_{txt}L_{txt} + \lambda_{face}L_{face} \tag{5}$$

where $\lambda_{txt}$ and $\lambda_{face}$ are the weights for balancing the losses.

### 3.3 Semantic reasoning about human faces

After being trained as described in the previous section, FaceGPT becomes a general model that can estimate 3D facial expressions from single images, create facial features based on detailed descriptions, and participate in question-and-answer dialogues. Most remarkably, even without directly training the model for linking 3D facial attributes to subtle phrases that do not directly contain specific facial traits, our model shows a zero-shot ability to reason about human faces from descriptions that contain emotions or general descriptions of everyday situations. This means the model can combine reasoning and world knowledge with the 3D facial representation. Similar as in ChatPose Feng et al. (2024), our aim is to highlight these emerging capabilities by introducing the task of *speculative face generation*, which focuses on the model's ability to reason about 3D human faces from speculative queries.

**Speculative Face Generation.** In this task, rather than providing explicit facial descriptions that directly detail the shape and texture of features, we pose speculative queries related to a person's emotional state or general everyday situations. The model is then tasked to infer a plausible 3D face, based on the assumption that the person is experiencing the described situation or emotion. For instance, a user might say, "Predict the face of a person who is excited about a surprise party." Answering such queries requires an understanding of broader concepts like "excitement" and the ability to deduce the appropriate facial features, followed by generating the relevant 3D facial parameters. To build an evaluation dataset, we draw from the CelebAText dataset Sun et al. (2021) for facial descriptions. We then use GPT-4 to rephrase these descriptions into questions about the emotions tied to each expression, resulting in a total of 444 responses, with 64 examples selected for evaluation. These responses undergo manual review and corrections in order to remove any direct descriptions of facial features. We further estimate the 3DMM face parameters of every test image using a state-of-the-art face autoencoder Li et al. (2023), resulting in paired data with speculative descriptions and corresponding 3D face parameters. More details can be found in supplementary.

## 4 Experiments

### 4.1 Training Data

**Text-to-Face Data.** The text-to-face dataset comprises pairs of text descriptions and face images, facilitating the development of mappings by VLM between textual descriptions and

3DMM parameters of faces. During training, only text is taken as input and the corresponding face images are only used for loss computation. Given the absence of publicly available datasets linking text descriptions to 3D face meshes and existing VLMs like LLaVA present powerful Visual Question Answering(VQA) ability, we rely on pre-trained VLMs to generate textual descriptions for face images. We employ following templates to guide the learning process in VLMs: `"USER: {description}, can you give the 3DMM parameters of this person. ASSISTANT: Sure, it is <FACE>."`, where `{description}` is the text description for faces. We utilize high-quality face images from CelebA-HQ (Karras et al., 2018) and use LLaVA (Liu et al., 2023a) to produce explicit text descriptions for dataset construction.

**Image-to-Face Data.** Image-to-face reconstruction data is composed of only human face images. The face images will be formatted with a template like `"USER: <IMAGE> Can you give the 3DMM parameters of this person. ASSISTANT: Sure, it is <FACE>."`. To enhance the diversity and relevance of the conversations centered around human face images, we also generate face-centric conversations for each face image. This approach enriches the contextual understanding of the VLM concerning the newly introduced token `<FACE>`. We adopt the CelebA-HQ trainset as the image to face reconstruction dataset.

**Multimodal Instruction-Following Data.** This data is general-purpose VQA data, and it is used to preserve the VLM's ability of understanding a user's instructions. Following LLaVA v1.5, We use LLaVA-v1.5-mix665k as multimodal instruction following data.

## 4.2 EXPERIMENTAL SETTINGS

**Network Architecture.** We build our model on Large Vision Language Model LLaVA-1.5-7B (Liu et al., 2023a) with CLIP-ViT-L-336px as vision encoder and Vicuna v1.5 as the LLM backbone. LoRA (Hu et al., 2022) is applied to efficiently fine-tune the VLM. An MLP head with GeLU activations (Hendrycks & Gimpel, 2016) and channels $[5120, 5120, 257]$ is appended to the last layer of the VLM to predict the 3D human face parameters.

**3D Face Model.** We use the Basel Face Model (BFM) 2017 Gerig et al. (2018) as the 3D face model. The face is parameterized as the semantic code vector $\theta = [\alpha, \delta, \gamma, \phi, c] \in \mathbb{R}^{257}$ in (Tewari et al., 2017), which includes 3D shape parameters $\alpha \in \mathbb{R}^{80}$, facial expression parameters $\delta \in \mathbb{R}^{64}$, texture of 3DMM $\gamma \in \mathbb{R}^{80}$, illumination $\phi \in \mathbb{R}^{27}$ and camera parameters $c \in \mathbb{R}^{6}$.

**LLaVa-Key baseline.** As there are no VLM-based methods that can perform 3D face reconstruction, we introduce the LLaVa-Key. The model is finetuned to predict facial landmarks in the format of pure text given either images or textual description of human faces as input . For each input, gradient-based optimization is applied to these predicted landmarks to fit the 3DMM using Eq. (3) to obtain a 3D face reconstruction. It is important to note that LLaVa-Key employs test-time fine-tuning, making it an inherently different baseline compared to all other feed-forward methods. Nevertheless, LLaVA-Key is useful to show the effect of a directly using the token space of VLMs to encode 3D information.

**Implementation Details.** The training uses 8 NVIDIA 48G A40 GPUs. We utilize deepspeed (Rasley et al., 2020) engine and ZeRO optimizer (Rajbhandari et al., 2020) for efficient training. We use AdamW (Loshchilov & Hutter, 2019) optimizer with learning rate and weight decay set to $2e-5$ and 0, respectively. We also follow the standard setting of using a WarmupDecayLR as the learning rate scheduler, where the warm-up iterations are set to 100. The weights of the text generation loss $\lambda_{txt}$ and the face reconstruction loss $\lambda_{face}$ are set to 1.0 and 0.1, respectively. Following (Li et al., 2023), those of the pixel loss $\lambda_{pixel}$, the perceptual loss $\lambda_{per}$, the landmark loss $\lambda_{LM}$, and the regularization loss $\lambda_{reg}$ are set to 0.5, 0.25, 5e-4 and 0.1, respectively. The landmarks of human faces are obtained with (Bulat & Tzimiropoulos, 2017) and pre-trained ArcFace (Deng et al., 2019a) is used to compute perceptual loss. The batch size is set to 8 per GPU and gradient accumulation step is set to 4.

**Evaluation Metrics.** For text-based face generation task, we compare the estimated 3D human face with the ground truth mesh to measure the Chamfer Distance(CD), Complete Rate(CR) and Relative Face Recognition Rate(RFRR) following Describe3D (Wu et al., 2023a). Both Chamfer Distance and Complete Rate would be used to reflect the accuracy of 3D meshes. And Relative Face Recognition Rate can measure the identity similarity of the textured 3D face rendering. We also additionally perform a user study for speculative face generation task as it is a difficult task requiring common sense of human. For image-based face reconstruction task, we measure the L2 photometric error in RGB space and the L2 landmark error to reflect the quality of 3D face's eometry and texture.

This person appears to have just received **a pleasant surprise**, like unexpectedly running into an old friend at a social gathering.

This person seems to be listening to a melancholic piece of music, **lost in thoughts and emotions** stirred by the melody.

This person is **shouting angrily** at a referee, reacting to a controversial call during a crucial sports match.

Implicit description   Describe3D   LLaVA-Key   FaceGPT

Figure 3: Qualitative results for speculative face generation. The abstract concepts and human activities in the descriptions are highlighted with bold text. FaceGPT presents a better capability in understanding the abstract concept and human activities compared to other methods.

Table 1: Quantitative results for speculative face generation. FaceGPT achieves a significant advantage in terms of the predicted 3D mesh's accuracy and user preference compared to other methods.

| Method | Unsupervised | CD $\downarrow$ | CR (%) $\uparrow$ | RFRR $\uparrow$ | User Study (%) $\uparrow$ |
|---|---|---|---|---|---|
| Describe3D | ✗ | 153.1 | 25.6 | 14.0 | 11.2 |
| LLaVA-Key | ✓ | 38.5 | 68.1 | 40.0 | 31.5 |
| FaceGPT | ✓ | **11.5** | **83.6** | **64.0** | **57.2** |

The instruction following ability is measured with GPT-assisted evaluation as described in Liu et al. (2023b), which queries GPT4 to obtain the grading of generated responses.

**User Study.** For evaluating the quality of speculative face generation task, we also perform a user study with 23 volunteers. Each volunteer will be presented with 20 questions from SPG benchmark and each question consists of an implicit description with the visual results generated by different methods given this description as input. For each question, the volunteer will be asked to utilize their understanding about the abstract descriptions and to select the result which best matches the implicit descriptions. The ratio of the number of times a specific method is selected to the total number of questions will be reported as the result.

### 4.3 SPECULATIVE FACE GENERATION

In this section, we evaluate FaceGPT's zero-shot capabilities at speculative face generation. We use the same template in Text-to-Face Data to query model and replace the {description} with the implicit description in the benchmark. For a fair comparison, we only select the face region of each method during the evaluation and align the 3D point clouds with the Iterative Closest Point (ICP) method before evaluation. The results are presented in Table 1 quantitatively and in Figure 3 qualitatively. We can observe that FaceGPT has significant advantages compared to Describe3D and the baseline LLaVA-Key method in terms of the quality of 3D shape and in terms of preference ratings in the user study. Interestingly, our baseline LLaVA-Key also outperforms Describe3D which requires a CLIP model and 3D supervision during training. In summary, FaceGPT develops a common sense about human faces that enables it to infer facial features from abstract and indirect descriptions.

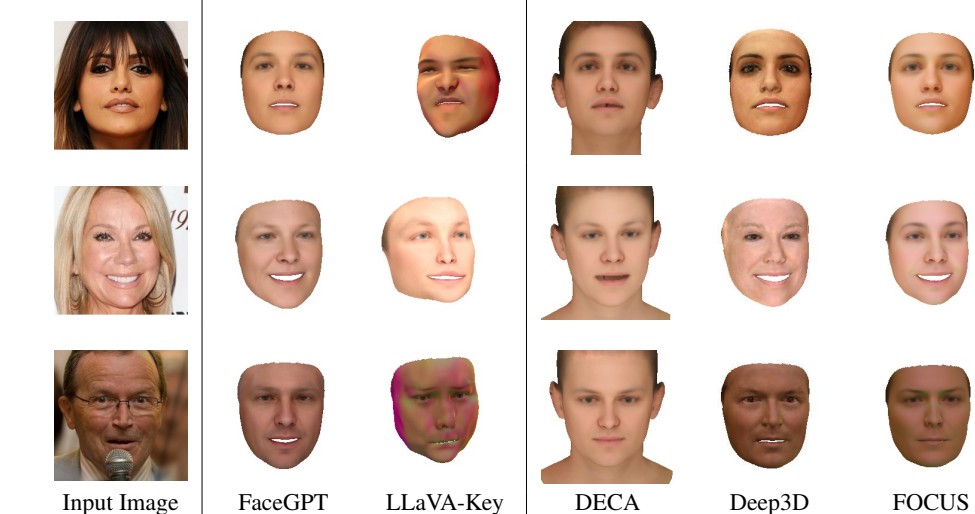

Input Image    FaceGPT    LLaVA-Key    DECA    Deep3D    FOCUS

Figure 4: Qualitative results for 3D face reconstruction. Our approach allows for the regression of pose, shape, expression, skin reflectance, and illumination from a single monocular image, achieving quality comparable to recent state-of-the-art methods.

## 4.4 EXPLICIT TEXT-BASED 3D FACE GENERATION

Table 2 shows the results of FaceGPT at 3D face reconstruction with explicit text description. As there are no public unsupervised methods available that perform this task, we compare FaceGPT to a supervised method Describe3D and our LLaVA-Key baseline. LLaVA-Key would fine-tune LLaVA to predict the 2D coordinates of face landmarks given text description as input and an optimization-based method is utilized to fit a 3DMM on predicted landmarks.

Like in speculative face generation benchmark, We also observe the significant advantage that FaceGPT achieves over other methods. Despite being trained in a self-supervised manner, FaceGPT can achieve faithful text-based 3D face reconstructions. These results demonstrate embedding 3D knowledge about our world into a VLM is in principle possible without detailed human annotations, hence demonstrating the large potential of combining VLM's with vision-as-inverse graphics for self-supervised learning. The qualitative results can be found in the supplementary.

Table 2: Performance of explicit text-based 3D face reconstruction. The evaluation is performed on shape and expression by comparing pseudo ground truth produced by FOCUS method.

| Method | CD ↓ | CR ↑ | RFRR ↑ |
|---|---|---|---|
| Describe3D | 96.88 | 29.3 | 16.27 |
| LLaVA-Key | 16.89 | 71.8 | 42.38 |
| FaceGPT | 7.28 | 91.7 | 64.88 |

## 4.5 IMAGE-BASED 3D FACE RECONSTRUCTION

We evaluate FaceGPT on image-based 3D face reconstruction and compare it with SOTA specialized methods for unsupervised monocular face reconstruction and a VLM-based baseline LLaVa-Key in Table 3.

Moreover, Fig. 4 shows a qualitative comparison of FaceGPT and all baselines at the classic task of 3D face reconstruction from a single image. As reported in prior works for VLM-based segmentation Lai et al. (2024) and human pose estimation Feng et al. (2024), our FaceGPT model does, as expected, not reach the state-of-the-art performance of specialized models. However,

Table 3: Performance at classical monocular 3D face reconstruction.

| Method | photo ↓ | keypoint ↓ |
|---|---|---|
| DECA | 0.216 | 5.2px |
| Deep3D | 0.073 | 3.2px |
| FOCUS | 0.077 | 2.2px |
| LLaVa-Key | 0.110 | 14.0px |
| FaceGPT | 0.103 | 3.0px |

Table 4: GPT4-Assisted Evaluation on instruction-following capability. "Conv", "Details", and "Complex" correspond to three types of questions(conversation, detailed description, complex reasoning) produced by LLaVA's data generation pipeline. GPT4 will be prompted to evaluate the answers from different models along with the ground truth answer produced by text-only GPT4 (gpt-4-0613). It would then give a score for each answer with an explanation.

|  | Conv | Detail | Complex | All |
|---|---|---|---|---|
| ChatPose (Feng et al., 2024) | 74.5 | 81.0 | 93.3 | 82.9 |
| LLaVA-V1.5-13B (Liu et al., 2023a) | 80.4 | 81.4 | 90.9 | 84.2 |
| LLaVA-V1.5-7B (Liu et al., 2023b) | 79.9 | 77.6 | 92.4 | 83.4 |
| FaceGPT | 79.6 | 81.5 | 92.6 | 84.6 |

Table 5: Influence of face-centric conversation generation

| Method | face convs | photo | keypoint | Conv | Detail | Complex | All |
|---|---|---|---|---|---|---|---|
| LLaVA-1.5-7B | ✗ | - | - | 79.9 | 77.6 | 92.4 | 83.4 |
| FaceGPT | ✗ | 0.110 | 3.2px | 78.4 | 80.8 | 89.0 | 82.7 |
| FaceGPT | ✓ | 0.103 | 3.0px | 79.6 | 81.5 | 92.6 | 84.6 |

we note that FaceGPT has a frozen vision encoder to preserve the generalist behavior, whereas all baseline models have a fine-tuned task-specific backbone. Moreover, our model does outperform DECA, which is a highly competitive and widely applied baseline model Zheng et al. (2023). When compared to the VLM baseline model, FaceGPT achieves large improvements, highlighting the benefit of embedding the 3DMM parameters directly in the token space of the VLM.

### 4.6 GPT-ASSISTED EVALUATION

Table 4 shows that FaceGPT preserves the ability of instruction following by following LLaVA's evaluation (Liu et al., 2023b) protocol, using GPT4-Assisted evaluation on LLaVA-Bench (COCO). FaceGPT compares favorably to LLaVA-v1.5-7B and reaches similar or better performance in the benchmark compared to VLMs with more parameters. This performance advantage can be attributed to the specialized face training dataset and the usage of our face-centric conversation. These elements enhance FaceGPT's proficiency in interpreting face-related language-based instructions, improving its overall effectiveness in relevant tasks.

### 4.7 ABLATION STUDY

**Influence of face-centric conversation generation.** To prove the necessity of our face-centric conversation generation strategy, we train a model only using a simple single-turn conversation template for face images, which is a common strategy used in the VLM-based image understanding works like LISA and ChatPose. The comparison results are demonstrated in Table 5. We observe that face-centric conversations help a lot in improving model's ability in performing detailed description and complex reasoning. The face-centric conversations only have a small effect on the face reconstruction task, which is expected, as the generated conversations do not contain information about the 3DMM parameters of faces.

## 5 CONCLUSION

FaceGPT is the first self-supervised learning framework for Large Vision-Language Models to reason about 3D human faces. We show that VLMs can learn to predict detailed 3D human faces from not only images, but also from textual inputs, in a fully self-supervised manner via inverse rendering. As a generalist model, FaceGPT achieves strong results across various tasks, including text-based face generation, traditional 3D face reconstruction, visual instruction following. FaceGPT also presents impressive ability to infer 3D faces from abstract and indirect text descriptions. We believe our work also has general implications beyond face analytics, as it points towards a way forward to enable large multi-modal language-models to reason about our 3D world without supervision.

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

# A APPENDIX

## A.1 SPECULATIVE FACE GENERATION BENCHMARK

> As an AI visual assistant specializing in human face analysis, your task is to infer possible activities, events, and emotional states a person might be experiencing based on a detailed visual and textual description of their face. Your goal is to focus on the high-level emotional context and plausible scenarios that this person could be engaging in, rather than anatomical details. For each facial description, consider the following key questions before formulating your response:
>
> 1. What emotion is the person likely experiencing? How does this emotion differ from a typical representation of this feeling?
> 2. What specific activity might this individual be participating in, based on their emotional state and facial expression?
> 3. What event could have triggered the current emotional expression or facial state?
> 4. Could the individual be engaged in other parallel activities that are influencing their expression?
>
> Once you've considered these questions, craft 5 distinct facial descriptions, each beginning with "This person," followed by one or two sentences that clearly suggest a plausible activity or situation. Ensure that each description provides a rich context that allows the user to imagine and even replicate the facial expression if they were in that situation. Avoid vague or general terms, and be as diverse as possible in your interpretations.
>
> Example answers and face descriptions:
> Answer to the questions:
> 1. The individual appears to be deeply worried, but there's an undertone of surprise not typical of worry alone.
> 2. The individual could be reviewing a critical work email and is taken aback by unexpected information.
> 3. It seems like the individual just encountered an unexpected delay for an important event.
> 4. Additionally, the person might be contemplating a difficult decision while trying to process sudden news.
>
> Facial descriptions:
> 1. This person looks like they have just received news that their flight has been canceled at the last minute.
> 2. This person is struggling to concentrate during a crucial work presentation, trying to mask their frustration.
> 3. This person might be reading an intense plot twist in a book, causing both confusion and intrigue.
> 4. This person seems to be on the verge of delivering uncomfortable feedback to a colleague.
> 5. This person appears to be caught off guard in a meeting, unexpectedly asked to answer a difficult question.

Figure 5: Prompts for querying GPT-4V to convert explicit text descriptions and facial images into implicit descriptions

Due to the lack of public datasets and benchmarks that provide implicit facial texts paired with corresponding 3D faces, , we choose to construct the Speculative Face Generation Benchmark with the help of powerful GPT4-V OpenAI et al. (2024). Inspired by Feng et al. (2024), we extract face images and human anotated facial descriptions to GPT4-V to generate implicit facial descriptions. Specifically, we select 74 high-quality facial images from CelebAHQ dataset Karras et al. (2018) and their explicit descriptions from CelebAText dataset Sun et al. (2021). Using the prompt detailed in Figure 5, we instruct the GPT4-V to analyze our provided images and detailed human descriptions and generate appropriate implicit descriptions to reflect the possible activities and the emotion

states associated with the faces. For each case, we ask GPT4-V to produce five potential implicit descriptions. We then fed these five candidate descriptions, along with the corresponding facial image, back into GPT-4V and requested it to select the implicit description that best matches the image. This process yielded a total of 444 responses from GPT-4V and 74 candidate text-face pairs. After manual verification, we finalized 64 test cases for our Speculative Face Generation benchmark.

## A.2 IMPLEMENTATION DETAILS ON LLaVA-KEY BASELINE

For Image-based 3D Face Reconstruction and text-based 3D Face Reconstruction, we develop a baseline based on VLM called LLaVA-Key where the human face is represented as text. Following Feng et al. (2024), we represent the human face through the 68 landmarks on 2D images as landmarks tracks the locations of eyes, nose, mouth and so on, which also contains rich information about faces. We utilize templates like `"USER: <IMAGE> Please estimate the 68 facial landmarks coordinates. The output format should be Jawline-1:(x1,y1),Jawline-2:(x2,y2),... ASSISTANT: The detected landmarks are Jawline-0:(41, 88),Jawline-1:(43, 107), ...,Inner Lip-67:(101, 159)."` for image-to-face data and templates like `"USER: There is a person with the following description:{description} Please estimate the 68 facial landmarks coordinates. The output format should be Jawline-1:(x1,y1),Jawline-2:(x2,y2),... ASSISTANT: The detected landmarks are Jawline-0:(41, 88),Jawline-1:(43, 107), ...,Inner Lip-67:(101, 159)."` for text-to-face data where `{description}` would be replaces with text description for faces generated by LLaVA. And we use an off-the-shelf face detector to provide the coordinates landmarks for each face. LLaVA model is finetuned to produce 2D coordinates of 68 faicl landmarks with the formatted data. After finetuning, LLaVA would be prompted to predict the landmarks in the test set and an optimization-based method is used to fit 3DMM for the estimated landmarks. For image-to-face data, we utilize $loss_{face}$ defined in 3 to optimize the 3DMM parameters on estimated landmarks and input image. For text-to-face data, we only utilize the landmark loss and regularization defined in 3.2 for optimization. Experiments reflect that representing human face as a new modality outperform naively encoding human face in language.

## A.3 FACE-CENTRIC CONVERSATION GENERATION

To enrich the diversity of question-answer pairs for image-to-face data, we inqury LLaVA with the face image and the questions listed in 6 and collect the answers from LLaVA to construct face-centric conversations for each face image. During training, we would randomly pick question-answer pairs from these generated conversation and fuse the conversation with task-static template presented in 4.1 as the text of training data.

Table 6: The list of instructions for constructing face-centric conversations.

- "How is the person's hair styled?"
- "What colors dominate this image?"
- "Based on the attire and styling, can you infer anything about the event or occasion for this photo?"
- "Can you describe the person's expression?"
- "Is there any indication of where this person might be?"
- "What is the person wearing?"

## A.4 TEXT-TO-FACE DATA GENERATION

As there is no public text-to-3DMM data available, we propose a way to utilize the powerful pretrained VLM to build a connection between face image with textual description. Specifically, we design

a template to inqury pretrained LLaVA to output detailed description about human faces, which is presented as following:

> Analyze the image and generate a detailed textual description of the human face it contains. Focus on the following aspects:
>
> 1. Face Shape: Description of the jawline shape (e.g., square, round, oval, heart-shaped). Forehead size and shape (e.g., wide, narrow, rounded). Cheekbone structure (e.g., high, low, prominent).
> 2. Face Expression: Eyebrows (e.g., arched, straight, furrowed). Eyes (e.g., wide open, squinting, normal). Mouth (e.g., smiling, frowning, neutral). Additional details if any specific expression is featured (e.g., wrinkling of forehead, dimples).
> 3. Face Color: Skin tone (e.g., fair, olive, dark, light). Any distinct color features such as freckles, rosiness, tan lines. Makeup if applicable (e.g., lipstick shade, eyeshadow color).
> 4. Face Lighting: Direction of the light source (e.g., frontal, side, backlit). Intensity of the light (e.g., soft, harsh, moderate). Shadows observed on the face (specify areas such as under eyes, neck).
>
> 5. Pose of Head: Mention the orientation of the head (e.g., facing forward, tilted to the side, looking upwards).
> Please give a response starting with 'He' or 'She'.

Figure 6: Prompts for querying LLaVA to generate explicit text descriptions on face images

The question covers many requirement on detailed description for many facial attributes and the response from pretrained LLaVA would be collected as the description for the face image and the 3DMM parameter of the corresponding face. Then we utilize these generated description with unsupervised face loss described in 3 to guide the model optimization.

### A.5 EVALUATION METRICS

For text-based face generation tasks, we generally use three metrics to measure the quality of the predicted 3DMM: Chamfer Distance(CD), Complete Rate(CR), and Relative Face Recognition Rate(RFRR). We will elaborate about how to compute each metric below:

- Chamfer Distance(CD): CD measures the similarity between two sets of point clouds. Given the predicted 3D mesh $M_p$ and the ground truth 3D mesh $M_g$, Chamfer Distance is defined as:

$$CD(M_p, M_g) = \sum_{a \in M_p} \min_{b \in M_g} \|a - b\|_2 + \sum_{b \in M_g} \min_{a \in M_p} \|a - b\|_2, \quad (6)$$

- Complete Rate(CR): CR is used to evaluate the completeness of a 3D point cloud, which can be formulated as the ratio between matched points and all points. The mathematical formulation of CR can be defined as:

$$CR = \frac{P_0}{P} \quad (7)$$

  $P_0$ is the number of points with CD value less than 10mm and $P$ is the number of all points.
- Relative Face Recognition Rate(RFRR): RFRR measures how well the reconstructed 3D face preserves identity information in comparison to the original face. For our case, we follow Wu et al. (2023a) and render the predicted mesh and the ground truth with the same pose and the lighting. Then we compute the cosine similarity in the feature space of ArcFace Deng et al. (2019a)to measure the idnetity preservation.

For image-based face reconstruction task, we measure the L2 photometric error and the L2 landmark error by comparing the rendered 3D face and the ground truth 2D images as they can directly reflect the quality of geometry and texture for the estimates meshes.

Table 7: Influence of LLM towards human face understanding

| Method | photo | keypoint |
|---|---|---|
| CLIP ViT + MLP | 0.133 | 7.6px |
| FaceGPT | 0.103 | 3.0px |

He has a square jawline and a prominent cheekbone structure. His forehead is narrow and his nose is of average size. His mouth is wide with thick lips that are slightly curved. His eyes are wide open, and he has arched eyebrows. His skin tone is fair, and he has a five o'clock shadow. The lighting is frontal, and the intensity of the light is moderate. The shadows are visible under his eyes and on his neck. He is facing forward.

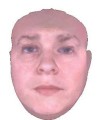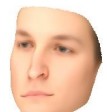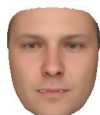

He has a smiling face with a wide mouth and thick lips. His eyes are wide open and his eyebrows are arched. His skin tone is olive and he has a prominent cheekbone structure. The lighting is frontal and the intensity is moderate. The man is facing forward with a smile on his face.

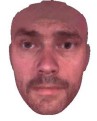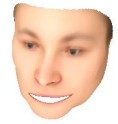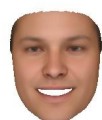

He has a square jawline, a wide forehead, and prominent cheekbones. His nose is large and straight, and his mouth is wide with thick lips. He is smiling in the image, showing off his teeth. His skin tone is dark, and he appears to be well-groomed. The lighting in the image is frontal, with a soft intensity, casting a gentle shadow under his eyes. The head is positioned facing forward.

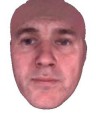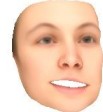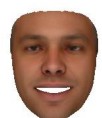

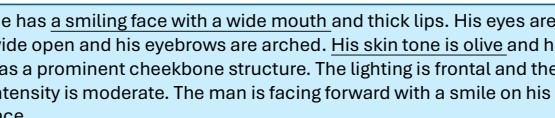

Explicit text description      Describe3D    LLaVA-Key    FaceGPT

Figure 7: We present some examples for explicit text-based face generation tasks. While some methods may overlook specific underlined details in the descriptions, FaceGPT consistently produces 3D faces with superior semantic alignment to the provided text.

- Photometric error: Photometric error can measure the quality of geometry and texture for 3D face estimation jointly. The error $E_{photo}$ can be formulated as:

$$E_{photo} = ||A \odot (x_{img} - \hat{y}_{rec})||_2 \qquad (8)$$

$A$ is the skin region in the ground truth face image.

- Landmark error: Landmark error can measure the quality of the geometry of 3D face and the accuracy of the estimated camera pose. The error $E_{lm}$ can be formulated as:

$$E_{lm} = ||LM_{img} - LM_{rec}||_2 \qquad (9)$$

## A.6 MORE RESULTS

### A.6.1 INFLUENCE OF LLM ON FACE UNDERSTANDING

Classical methods mainly rely on a vision encoder for regressing the 3DMM face parameters. In contrast, LLaVA utilizes a combination of frozen vision encoder and an LLM to perceive information from images. To study the effect of the LLM head, we train a baseline model with the vision encoder of LLaVA and an MLP to predict the human face parameters directly. The results are presented in Table 7 and show that the additional LLM helps in predicting better face parameters based on the visual representations from the frozen encoder.

### A.6.2 COMPARISON BETWEEN SUPERVISED FACE LOSS AND UNSUPERVISED FACE LOSS

Recent VLM-based works on segmentation and pose estimation generally rely on supervised learning with ground truth data or pseudo ground truth data. To study the influence of supervision on the model's capability, we compare the model trained with supervised 3DMM losses and self-supervised

Table 8: Comparison when learning with supervised and unsupervised losses.

| Method | photo. | keyp. |
|---|---|---|
| FaceGPT(sup) | 0.092 | 2.1px |
| FaceGPT(unsup) | 0.103 | 3.0px |

2D face losses. For the supervised 3DMM loss, we extract the 3DMM parameters on the same CelebA-HQ trainset with the state-of-the-art face reconstruction method FOCUS (Li et al., 2023). L1 loss is used for optimization on 3DMM ground truth. We observe in Table 8, that the model utilizing a supervised 3DMM loss outperforms its unsupervised counterpart, indicating that the performance ceiling is not reached yet and improvements on the self-supervised training could potentially lead to further performance gains.

### A.6.3 QUANTITATIVE RESULTS FOR EXPLICIT TEXT-BASED FACE GENERATION

We present visual results for explicit text-based face generation in Figure 7. Compared to other methods, the output of FaceGPT are generally better aligned to the text descriptions.

