# OpenReview forum: "FaceGPT: Self-supervised Learning to Chat about 3D Human Faces"
_ICLR.cc/2025/Conference — ICLR 2025 Conference Withdrawn Submission_

### Official Review · Reviewer_TB1y · 2024-10-25

**Soundness:** 2
**Presentation:** 2
**Contribution:** 2
**Rating:** 3
**Confidence:** 4

**Summary:**

This paper fine-tunes a VLM (LLaVA-1.5-7B), unifying image-based 3D face reconstruction and language-based 3D face generation. Although the paper claims to be a self-supervised learning framework, the actual content indicates the use of supervision signals provided by off-the-shelf face reconstruction methods and VLM. It is effectively a supervised learning approach! Its loss function comprises two parts: the loss function for generating 3DMM output and the loss function for instruction-tuning.

**Strengths:**

This paper presents a valuable topic: constructing a unified model for generating 3D faces from both images and texts. Specifically, speculative face generation holds significant value in fields such as criminal tracking. The experiments also demonstrate the effectiveness of the constructed model in speculative face generation, explicit text-based 3D face generation, and image-based 3D face reconstruction.

**Weaknesses:**

The core idea of self-supervised learning is to set up proxy tasks that allow the model to train and capture the intrinsic structure and features of the data in the process. Although the paper claims to use a self-supervised learning framework, there seems to be some deviation from the conventional definition of self-supervised learning.
Based on the details of training and data construction in the paper, the method employed appears to be a straightforward supervised learning approach, similar to the instruction-based fine-tuning executed in papers like LLaVA. From the content on lines 193 and 236, it seems that the authors believe an algorithm can be considered self-supervised as long as it does not use manually annotated data. This perspective might reflect a different interpretation of the concept of self-supervised learning.
Although the paper does not introduce manually annotated data, it utilizes off-the-shelf face reconstruction methods and VLMs to construct 3DMM data and textual descriptions from 2D face images. This effectively means that off-the-shelf models are being used to provide supervisory signals for training.

**Questions:**

1. What is the definition of self-supervised learning according to the authors, and how does it differ from conventional interpretations?
2. How does the paper's approach to training and data construction align with or deviate from traditional self-supervised learning methods?
3. Can the utilization of off-the-shelf models for generating 3DMM data and textual descriptions from 2D face images be considered a form of supervisory signal?

---

### Official Review · Reviewer_PMqq · 2024-11-03

**Soundness:** 2
**Presentation:** 2
**Contribution:** 2
**Rating:** 5
**Confidence:** 5

**Summary:**

The paper proposed a novel framework to use large VLMs to reason about 3D human faces from text/images by embedding the 3DMM face parameters into the token space. The framework is train in a self-supervised manner using image-based reconstruction and differentiable rendering. The authors claim that the proposed framework is able to leverage the existing work knowledge in VLMs to achieve semantic reasoning capabilities for 3D face generation.

**Strengths:**

1. The paper is well written and easy to follow.
2. The paper proposed a framework that can leverage large VLMs to generate 3D faces from natural description of emotions.
3. The framework doesn't require any coupled text and 3D face data.
4. The framework achieved good 3D face reconstruction results.

**Weaknesses:**

1. Insufficient Justification for Using VLMs:
The paper does not provide adequate justification for employing Visual Language Models (VLMs) in the 3D face synthesis task. The outcomes presented could potentially be replicated by many existing methods if trained under a conditional framework incorporating a CLIP text encoder along with detailed textual descriptions.

2. Subpar Quality of Generated Faces:
The quality of the generated faces significantly lags behind the current state-of-the-art face reconstruction methods. This is primarily attributed to the use of the outdated 3DMM model—BFM—which yields a linear texture and a very coarse mesh, limiting the detail and realism of the synthesized faces.

3. Lack of Standard Benchmarking:
It would be beneficial to evaluate the performance of this framework against standard 3D face reconstruction benchmarks, such as the Now and Realy3D datasets. Additionally, an analysis of the quality of the reconstructed mesh would provide a clearer picture of the framework's capabilities.

**Questions:**

1. Choice of 3DMM Model:
Why does the framework utilize an older 3DMM model like BFM instead of more recent models that can capture finer facial details?

2. Reasoning Capabilities of VLMs:
Is there empirical evidence to support that VLMs possess the reasoning capabilities to accurately interpret human faces? If not, why prefer this framework over specialized existing frameworks designed for such tasks?

3. Reliability of VLM Outputs:
The framework presupposes that the VLM will consistently output the <FACE> token when analyzing a face. Are there instances where the VLM fails to produce a <FACE> token even when expected?

4. Verification of VLM-Generated Descriptions:
Is there a method to verify the accuracy of the descriptions generated by the VLM? [Ref. Lines 274-276]

5. Training Methodology:
The approach of using descriptions generated from VLM to re-train the VLM for estimating 3DMM parameters appears circular, akin to using knowledge distillation within the same model. Is there a more effective method to accomplish this?

6. Contribution of VLM to the Framework:
To what extent does the VLM contribute to the overall framework's effectiveness? Could similar results be achieved using simpler language models or the CLIP text encoder alone? [Ref. Lines 299-300]

7. Necessity of Detailed Descriptions:
In scenarios such as "Predict the face of a person who is excited about a surprise party", it seems that a simple description of the expression (e.g., "excited") might suffice. If a human would be asked to draw/imagine a face with this description, there are pretty good chances they will simply draw/imagine a face with "excited" expression on it. The additional narrative appears redundant. Do language models require this excess information to generate accurate facial expressions? Why do we really need the accompanying redundant information simply to generate a face with "excited" expression. I made the same observation in the Fig.3 examples where the faces only convey the main expression like "surprise", "lost", or "angry".

8. Modeling complex expressions:
Could the authors demonstrate complex expressions or combinations of expressions that existing models fail to capture to show the effectiveness of this framework?

---

### Official Review · Reviewer_KkuK · 2024-11-03

**Soundness:** 2
**Presentation:** 2
**Contribution:** 2
**Rating:** 3
**Confidence:** 4

**Summary:**

This paper describes a method where a VLM is trained with LORA to be adapted for the task of 3D face reconstruction. The VLM is supposed to provide textual information describing the face and in the end of the text a "face" token is used to predict 3DMM face parameters.

**Strengths:**

The combination of VLMs with face-related tasks has not been explored in literature and in its current instantiation in this paper presents some amount of novelty.  Moreover, training the VLM with a face reconstruction training objective in a self-supervised manner bears some degree of novelty.

**Weaknesses:**

Unfortunately, I cannot grasp the motivation behind the proposed work as in the end of the end day it boils down how to fine-tune a VLM for 3D face reconstruction. But there are already several state-of-the-art methods of high accuracy for this task. Similarly there are already several methods for text-driven face generation. It's not clear if the proposed method is any better than methods tailored to these tasks. Importantly, these are vision tasks so it is unclear why a VLM is needed and what extra capabilities are brought into play by using for solving these tasks. The paper fails to demonstrate some newly introduced capability regarding the understanding of human faces that we have seen before. The speculative face generation task is poorly described and the evaluations do not make a lot of sense. This can be illustrated by looking at the results of Fig. 3. Clearly the method  has not really been trained successfully to produce high quality realistic faces corresponding to the textual descriptions used as input. Finally, even for face reconstruction the proposed method produces poor results as the visual results of Fig. 4 show.
Overall, the paper fails to show why VLM is needed for traditional 3D tasks, does not introduce any new capability and also fails to show decent results for the tasks it's evaluated for

**Questions:**

Please see weaknesses above. Moreover:

- Please provide a more comprehensive comparison with existing state-of-the-art methods for both 3D face reconstruction and text-driven face generation.
- What unique capabilities or insights does the language component provide that purely vision-based approaches lack? Please provide concrete examples of how the VLM enhances face understanding beyond existing methods.
- Do the authors believe that the generated faces of Fig. 3 accurately capture the input text descriptions?
- What's the dataset used for the evaluation of Table 3? Are you comparing with SOTA? As said the results of Fig. 4 don't look very compelling as there are many details missing from the reconstructed faces and the identity is not well preserved.

---

### Note · Authors · 2024-11-13

I have read and agree with the venue's withdrawal policy on behalf of myself and my co-authors.